# A DNA Sequence Based Polymer Model for Chromatin Folding

**DOI:** 10.3390/ijms22031328

**Published:** 2021-01-29

**Authors:** Rui Zhou, Yi Qin Gao

**Affiliations:** 1Biomedical Pioneering Innovation Center, Peking University, Beijing 100871, China; zhourui@pku.edu.cn; 2Beijing Advanced Innovation Center for Genomics, Peking University, Beijing 100871, China; 3College of Chemistry and Molecular Engineering, Peking University, Beijing 100871, China; 4Beijing National Laboratory for Molecular Sciences, Peking University, Beijing 100871, China; 5Shenzhen Bay Laboratory, 5F, No.9 Duxue Rd., Nanshan District, Shenzhen 518055, China

**Keywords:** chromatin, polymer model, genomic sequence

## Abstract

The recent development of sequencing technology and imaging methods has provided an unprecedented understanding of the inter-phase chromatin folding in mammalian nuclei. It was found that chromatin folds into topological-associated domains (TADs) of hundreds of kilo base pairs (kbps), and is further divided into spatially segregated compartments (A and B). The compartment B tends to be located near to the periphery or the nuclear center and interacts with other domains of compartments B, while compartment A tends to be located between compartment B and interacts inside the domains. These spatial domains are found to highly correlate with the mosaic CpG island (CGI) density. High CGI density corresponds to compartments A and small TADs, and vice versa. The variation of contact probability as a function of sequential distance roughly follows a power-law decay. Different chromosomes tend to segregate to occupy different chromosome territories. A model that can integrate these properties at multiple length scales and match many aspects is highly desired. Here, we report a DNA-sequence based coarse-grained block copolymer model that considers different interactions between blocks of different CGI density, interactions of TAD formation, as well as interactions between chromatin and the nuclear envelope. This model captures the various single-chromosome properties and partially reproduces the formation of chromosome territories.

## 1. Introduction

People have long been interested in the folding of DNA in the nucleus. The recent development of experimental techniques such as chromosome conformation capture (especially, Hi-C) [1] and super-resolution imaging techniques [2,3] has revealed an unprecedentedly detailed picture of the chromatin structure in the nucleus. It was found that the chromatin takes a hierarchical folding pattern [1,4,5,6,7]. At the scale of hundreds of kilo-base pairs (kbps), the topological associating domains (TADs) have been revealed, which are contiguous loop-like structures along the sequence. The interactions inside TADs are stronger than between TADs [4,5,6]. At the megabase scale, several characteristics of chromatin folding and spatial organization in the nucleus have also been identified. Firstly, the Hi-C technique reveals that the chromatin folds into two types of spatially segregated megabase compartments in the nucleus, the compartment A and the compartment B [1]. Segments belonging to the same type of compartment tend to interact with each other, while segments belonging to different types of compartments tend to separate spatially. Interactions between B compartments are stronger than those between A compartments [1]. However, the TADs in compartments B are large and lack strong interactions inside, while the TADs in compartments A are small and condensed [8], indicating that intra-compartment A interactions are stronger than intra-compartment B. Secondly, a power-law decay of the contact probability *P* between loci with the sequential distance *N*, say, *P*(*N*)~*N*^−^^α^, is observed [1,9,10]. However, the value of α varies with the genomic distance. At the scale of TADs (below several hundred kbps), α is usually smaller than one [1,11]. The value of α typically lies in the range between 1 and 1.3 at distances of several hundred kbps to about 7 Mbps [1,9,10]. At even larger genomic distances (7~20 Mbps), α decreases again to about 0.6~0.8 [1]. Thirdly, compartment B usually tends to occupy the nuclear periphery and the nuclear center, while compartments A occupy other spaces [1,3,12]. These results indicate that from the nuclear periphery to the nuclear center, the main component of the chromatin forms roughly a B-A-B distribution pattern, although in certain cells, such as rod cells, compartments A lie near the periphery and compartments B locate near the nuclear center [13,14]. It was also found that chromosomes tend to segregate and form chromosome territories [12].

Recent studies additionally revealed that chromatin folding properties are correlated with many other biological factors, such as genomic content, epigenetic marks, gene density and expression intensity [1,7,15,16]. At the megabase scale, compartments A mainly correspond to the euchromatin, which is gene rich, abundant in activating histone marks, low in DNA methylation level and actively transcribed. Compartments B mainly correspond to heterochromatin with characteristics opposite to compartments A [1]. It was also found that the organization of compartments B is well correlated with the lamina-associating domains (LADs, see below), which tend to interact with the nuclear lamina and populate near the nuclear periphery [1,5,17,18]. In our previous study we separated the mammalian genome into two kinds of domains based on the CpG island (CGI) density, namely the CGI-rich forest (F) domains and the CGI-poor prairie (P) domains [19]. These domains strongly correlate with compartment formation. F domains mostly colocalize with compartments A, while P domains mostly colocalize with compartments B [19]. Previously, we proposed that the establishment of F-F and P-P interactions results in a segregation between F domains and P domains, which leads to compartmentalization. Further studies are needed to improve our physical understanding of chromatin folding principles and test the feasibility of the proposed model.

The importance of lamina association was highlighted in chromatin spatial organization. Lamin proteins populate mainly near the inner-surface of the nucleus (the nuclear lamina), and the LADs are sequences interacting with these proteins. Single cell experiments showed that not all LADs bind to the nuclear lamina. A fraction of them detach from the nuclear lamina and condense into the nuclear center to form nucleolar associating domains [20,21]. Lamin proteins play an important role in the formation of LADs. Rod cells lack lamin B proteins and exhibit a reversed spatial organization compared to other cells, with compartments A being close to the nuclear periphery [13,14]. These studies combined with a recent report on the importance of heterochromatin in compartmentalization [3] imply that condensation of heterochromatin and lamina binding are both important driving forces in the formation of chromatin spatial organization.

Polymer modeling is widely employed to study the folding mechanism of chromatin. In these studies, the chromatin is typically modeled as a polymer chain, while the interactions are modeled by direct interactions between beads on the polymer [22,23,24,25] or mediated by proteins [26,27,28,29,30] simplified as free beads. Many characteristics of chromatin folding have been reproduced or revealed from these rather sophisticated model studies. For example, the string and binder switch (SBS) model of Barbieri et al. used free beads to mediate interactions between chromatin segments, and reproduced the scaling properties of the entire chromatin, topological domain formation, as well as looping out [26]. The loop extrusion model described a process that protein complexes slide along the chromatin, which can be halted by TAD boundaries [11,31]. This latter model successfully explains the formation of TADs. Furthermore, by combining the loop extrusion model with a block copolymer model, J. Nuebler et al. studied the interplay between the formation of TADs and compartments [32]. In recent studies, the block copolymer model and LAD formation have been combined to illustrate the spatial organization of chromatin. These models generally divide the chromatin into different kinds of blocks. Blocks of the same kind attract each other. For some blocks, attractions between blocks and nuclear periphery are applied to mimic the LAD formation. For example, M. Falk et al. considered three kinds of blocks, one kind of which could interact with the nuclear lamina [3]. Their results indicate that the LAD formation plays a crucial role in forming the spatial organization pattern for both normal cells and rod cells. M. Chiang et al. [33] and S. Sati et al. [34] used this type of model to study the formation of senescence-associated heterochromatin foci (SAHF). Though the criteria they used to divide blocks are different, their studies both indicate the important roles of the interactions between related heterochromatin domains and the nuclear lamina, as well as interactions between heterochromatin domains, in the formation of SAHF. Although significant variations do exist between different cell types, different (especially differentiated) cells share many common (and to a large extent conserved) features such as those in TAD [4,5] and compartment formation [19]. A model that provides a simple understanding of the general folding properties of chromatin, such as the formation of TADs and compartments, decay of the contact probability with the genomic distance, different spatial organization patterns of compartments A and B, and different intra- and inter-domain interaction profiles of compartments A and B, and in particular, the role of the DNA sequence, is still needed.

In this study, we established a coarse-grained polymer model taking into account the mosaic DNA sequence property as reported before [19], the TAD formation, as well as interactions between chromatin and nuclear lamina. We considered both single chromosome, as well as several chromosomes. Based on these models we characterized the reported chromatin folding properties at different scales and on many aspects, including TAD and compartment formation, spatial organization, and spatial separation between chromatins. We also studied the effects of parameters used in this model.

## 2. Results

### 2.1. General Properties of Chromatin Folding Are Reproduced by Our Model

A graphical illustration of our model is shown in Figure 1a. In our model, one chromosome is coarse-grained into a polymer with each bead representing 100 kbps. The information about F/P domains, TAD boundaries and LADs is then mapped to each bead. We applied attractive Lennard-Jones (LJ) potentials between each P-P, F-F and P-F bead pair of different intensity, marked as *ε*_PP_, *ε*_FF_, and *ε*_PF_, respectively. To form loops between adjacent TAD boundaries and for simplicity, we applied different harmonic potentials (*ε*_PB_ and *ε*_FB_) according to the genomic content of the TAD. The model chain is then placed into a spherical container representing the nucleus, and the chromosome occupies a volume fraction *φ* = 5% of the container volume. Between LADs and the nuclear envelope, an attractive LJ potential of intensity *ε*_LC_ is applied. We first present the simulation results on a single chromosome, human chromosome 10 (chr10), the F/P distribution and LAD distribution along genomic sequence of which are shown in Figure 1b. In this model, the radius of the container is 16.7*σ*_0_. A typical contact map generated from the simulation using *ε*_PP_ = 4.6*ε*_0_, *ε*_FF_ = 4.3*ε*_0_, *ε*_PF_ = 3.0*ε*_0_, *ε*_PB_ = 0.2*ε*_0_, *ε*_FB_ = 0.4*ε*_0_, *ε*_LC_ = 6.7*ε*_0_ for chr10 is shown in Figure 1c. A checkerboard-like pattern was obtained, consistent with compartmentalization in chromosomal 3D structure. TAD-like structures can also be seen in Figure 1c. Due to different interaction intensities between boundaries of TADs composed of mainly F domains (F-TADs) and TADs composed of mainly P domains (P-TADs), and different sizes of F-TADs (4.4 beads long in average) and P-TADs (6.6 beads long in average), different contact patterns are observed for F and P domains in the contact map. While the boundaries of P-TADs are sharp and clear, the boundaries of F-TADs are fuzzy and indistinct. To quantify these different patterns, we applied a windowed Fourier transform to the 5th diagonal of the contact matrix, with a window size 2 Mbps. The results of Fourier transform for windows containing mainly F and P domains are averaged, respectively, and shown in Figure 1c. The existence of two types of TADs is consistent with the Hi-C data [8].

We next calculated the contact probability *P* as a function of the genomic distance *N*. Generally, the contact probability *P* decays with the increase in *N* following a power law, *P*(*N*)~*N*^−^^α^. Our simulation shows that the value of α varies with genomic distance, which is about 0.8 for *N* < 700 kbps, ~1.3 for 700 kbps < *N* < 7 Mbps, and ~0.6 for *N* > 7 Mbps, as seen from Figure 1d. Such a decay pattern is in agreement with the Hi-C data up to 20 Mbps (Figure 1d). We further investigated the contact probability as a function of *N* for F and P domains, respectively, as shown in Figure 1e. At the scale of TADs (less than 1 Mbps), the contact probability for F domains is larger than that for P domains, indicating the effects of stronger interactions in the former than the latter in forming TADs. Since P-TADs are generally larger than F-TADs, a higher contact probability is observed for P domains than for F domains at around 1 Mbps. Due to stronger P–P interactions, the higher contact probability for P domains than F domains persists up to tens of Mbps, at distances longer than which no obvious difference between the two types of domains was observed. The ratio between the contact frequency for F and P domains as a function of genomic distance is now compared with experimental data (Figure 1f), showing a reasonable agreement except for a slight overestimation at the TAD scale. At the range of 7 Mbps to ~20 Mbps, the slope of the decay curve for both P and F domains becomes smaller compared to that in the range of 700 kbps to 7 Mbps, and the slope for P domains becomes obviously smaller than that for F domains. Such a decrease in slope for both domains results in a slower decay in the range of 7 to 20 Mbps compared to 700 kbps–7 Mbps in Figure 1d. The differences between the slope for F and P domains are consistent with the different intensities in P–P and F–F interactions (see below).

We next analyzed the radial distribution functions (RDFs) for F and P domains, respectively (Figure 1g). The RDF for P domains exhibits a double-peak profile, exhibiting a sharp peak near the nuclear periphery and a broad one near the center. The RDF for F domains, on the other hand, has only one peak, residing at an intermediate position. Therefore, the organization of the chromatin can be roughly characterized by a three-layered F-P-F spatial distribution as reported in Ref. [12].

To examine the inter-chromatin interactions, we next performed simulations on a system with two chromosomes, chr10 and chr14. The profiles of P domains and LADs of chr14 are shown in Figure 2a. All parameters are the same as those used in the single chromosome simulation discussed above (the radius of the container is changed to 19.8*σ*_0_ to maintain *φ* = 5%). From the calculation of the contact map (Figure 2b), we again observed the formation of TADs and compartments. Therefore, this latter simulation reproduces the features of single-chromatin folding. The slow–rapid–slow decay pattern of the contact probability as a function of sequential distance is also reproduced in this simulation of two chromosomes (Figure 2c). At the scale of TADs (300–700 kbps), both the 2-Chr and the 1-Chr simulation systems yield α~0.8, implying the main role of TAD formation at this scale. However, at larger scales, the α value given in the 2-Chr system becomes higher than that in 1-Chr simulation (for example, α = 1.37 in the former vs. α = 1.32 in the latter for 700 kbps < *N* < 7 Mbps) as shown in Figure 2d, indicating that the conformation of chr10 in the 2-Chr system is more open than that obtained from the single chromosome simulation. Furthermore, the P-F-P three-layered spatial organization pattern with a double-peak RDF pattern for P domains and a single-peak RDF pattern for F domains is also observed in the 2-Chr system (Figure 2c), though the density of P domains in the nuclear center decreases. When the two chromosomes were analyzed separately, as shown in Figure 2e,f, we found that the RDFs for chr10 and chr14 differ from each other. For example, the separation between the two peaks of the P domains for chr14 is more obvious than chr10, implying their difference in DNA sequences. Additionally, the density of P domains located near the center in chr14 is higher than that in chr10, which might result from the fact there are fewer LADs in the former than the latter (see below).

### 2.2. The Formation of Chromosome Territories Is Partially Reproduced

To examine the extent of chromosome mixing, we compared intra- and inter-chromatin interactions in the 2-Chr system. In our model, the average long-range (>5 Mbps) contact probability is 0.0088 for chr10 and 0.0130 for chr14, and the average contact probability between chr10 and chr14 is 0.0079. Therefore, chr10 and chr14 in the 2-Chr system show a slight tendency to separate. To further examine whether the DNA sequence difference plays a role in the spatial organization of different chromatins, we also used chr18 and chr19 as examples. These two chromosomes have very different sequential properties: chr18 is rich in P domains and LADs (Figure 3d), while chr19 is rich in F domains and lacks LADs (Figure 3e). We simulated a 3-Chr system containing chr14, chr18 and chr19, in which all parameters are the same as those used in the 2-Chr model as the amounts of beads are almost identical in these two models. The average long-range contact frequencies for chr14, chr18 and chr19 are 0.0122, 0.0101 and 0.0146, respectively. These intra-chromatin contact frequencies are again slightly higher than the average long-range contact frequency between chr14 and chr18 (0.0083) and that between chr14 and chr19 (0.0097), but are significantly higher than the average long-range contact frequency between chr18 and chr19 (0.0053). Therefore, the large sequence difference between chr18 and chr19 does result in a large extent of separation between them. The high separation tendency between chr18 and chr19 is also reflected from their very different RDFs. As shown in Figure 3b, P domains in chr18 exhibit very high tendency to populate near the periphery and very few beads of chr18 populates in the center, while chr19 is excluded from the periphery (Figure 3c). This high tendency of isolation between chr18 and chr19 is again in line with their large differences in sequence.

### 2.3. Bonding between TAD Boundaries Affects the Decay of Contact Probability with Genomic Distance

We also examined how different interaction parameters used in the models affect the simulation results. Firstly, we varied the bonding intensity between TAD boundaries to test how this affects the folding patterns. In this series of simulations, we fixed *ε*_FB_ to 0.4*ε*_0_, and changed *ε*_PB_ from 0.1*ε*_0_ to 0.4*ε*_0_. Such a change in bonding intensity was shown to strongly affect the α values, especially for *N* < 7 Mbps, as shown in Figure 4a. With the increase in *ε*_PB_, the decay in contact probability at the scale of TADs becomes less steep, with α = 0.86 at *ε*_PB_ = 0.1*ε*_0_ and α = 0.74 at *ε*_PB_ = 0.4*ε*_0_ for 300 kbps < *N* < 700 kbps. On the other hand, the decay becomes more rapid in the region 700 kbps < *N* < 7 Mbps, with α changing from 1.21 at *ε*_PB_ = 0.1*ε*_0_ to 1.37 at *ε*_PB_ = 0.4*ε*_0_. In contrast, the contact probability at long distances does not change significantly with *ε*_PB_. As for the RDF, all *ε*_PB_ values tested here result in a similar P-F-P three-layered spatial distribution (Figure A1), suggesting that interactions at the scale of TADs do not affect strongly the large-scale chromatin spatial distribution.

### 2.4. F–F, P–P Interactions and LAD Formation Mainly Affect Spatial Organization of Chromatin

The other important factors affecting chromatin organization, especially compartment formation, are the intra-domain interactions, characterized in the current model by *ε*_FF_ and *ε*_PP_. In a series of simulations, we fixed *ε*_PP_ to 4.6*ε*_0_, and systematically changed *ε*_FF_ from 4.1*ε*_0_ to 4.5*ε*_0_. The simulation results show that such changes in *ε*_FF_ have little effect on the relationship between the contact probability and the genomic distance (Figure 4b). Similarly, the changes in *ε*_PP_ from 4.4*ε*_0_ to 4.8*ε*_0_ (with *ε*_FF_ being fixed to 4.3*ε*_0_) also have a very small effect on how the contact probability varies with the genomic distance (Figure 4c). In contrast, the increase in *ε*_FF_ and(or) *ε*_PP_ does result in an increase in both P-P and F-F contact probability (Figure A2 and Figure A3). In addition, similar values of *ε*_FF_ and *ε*_PP_ also result in similar decay patterns of the contact probability of P domains and F domains as a function of the sequential distance at all distance ranges examined. As a result, *ε*_FF_ and *ε*_PP_ affect the relative intensities in F-F, P-P and F-P contact and thus the spatial organization (Figure 4d,e). With increasing *ε*_FF_ and decreasing *ε*_PP_, the P domains exhibit a higher tendency to populate near the nuclear periphery but a lower tendency to populate near the nuclear center. The observation that a large *ε*_PP_ facilitates P domains to condense in the nuclear center implies a competition between *ε*_PP_ and *ε*_LC_, as large values of *ε*_LC_ induce P domains to move towards the periphery.

We next examined the effects of the LAD-container interaction strength, *ε*_LC_, on chromatin folding. In the corresponding simulations, we fixed all other parameters except for *ε*_LC_ which takes the value of 6.5, 6.7, 6.9, or 7.2*ε*_0_. The decay pattern of contact probability is found to be insensitive to the change of *ε*_LC_ (Figure 5a). However, as expected, *ε*_LC_ significantly affects the radial distribution (Figure 5b). At large *ε*_LC_, LADs tend to be attached tightly near the periphery, and consistently one observes a high tendency for P domains to populate near the periphery. When a small *ε*_LC_ is used, LADs tend to detach from the periphery and condense in the nuclear center (Figure 5b), which is also noted in Refs. [3,33,34]. For an intermediate *ε*_LC_ (such as 6.7*ε*_0_), a P-F-P three-layered spatial organization is obtained, with more P domains residing near the nuclear periphery than at the center, as shown in Figure 5b.

To evaluate the effects of LAD pattern, we finally considered two more LAD samples of chr10, the profiles of which are shown in Figure 6a,b. The LADs in Figure 6a are of relative low population and concentrated, while the LADs in Figure 6b are more abundant and dispersed. We concentrated mainly on the RDF. As shown in Figure 6c, the P domains of the sample in Figure 6a condense into the nuclear center. In contrast, the P domains of the sample in Figure 6b are drawn to the periphery (Figure 6d), and neither P nor F domains exhibit a preference for the central location. These results indicate that the abundance and distribution of LADs strongly affect the spatial organization of the interphase chromatin.

## 3. Discussion

In this study, we tested how DNA-sequence based chromatin models can be used to reproduce general properties of chromatin folding, such as the formation of TADs and compartments, the decay pattern of the contact probability as a function of the genomic distance, and the spatial organization of inter-phase chromatin. When more than one chromatin is used in the simulation, we partially reproduced the formation of chromosome territories. We investigated the effects of a number of factors that might influence the folding of chromatin, including bonding interactions between TAD boundaries, the inter-block interactions, and the interactions between LADs and the nuclear envelope.

Our simulation reproduces the contact probability decay pattern, in agreement with Ref. [1] and many other samples [35,36], for genomic distances up to 20 Mbps. Generally, the decay follows a power law, *P*(*N*)~*N*^−^^α^. In short distances (<700 kbps), a slow decay is observed with α~0.8. At the intermediate region of 700 kbps~7 Mbps, the contact probability decays rapidly, with α generally being in the range 1.0~1.3. Finally, at long distances (>7 Mbps), a slow decay is again observed (α~0.6). At the short range, our simulation showed that the slow decay is due to the formation of TADs. TAD formation has been the focus of a number of models of different levels, such as the loop extrusion model [11,31] and the self-returning random walk model [37]. These models satisfactorily captured the TAD formation, and thus, the slow decay pattern in this distance range. The intermediate distance region was originally described by a fractal globule model, although it has also been reproduced by other models, such as the SBS model [26] and the dynamic-loop model [38]. The long-distance behavior of the decay curve has been reproduced by the SBS model [26,30]. In our model, the fast and then slow decay at distances longer than the TAD size appear to result from the heterogenous nature of the block copolymer. For a block copolymer with blocks of equal length, a periodically oscillating contact probability is observed as a function of the sequential distance [39]. Our model presented here reproduces the decay curve covering all three regions. In addition, we compared the decay pattern of the contact probability as a function of *N* for F and P domains, respectively. With stronger *ε*_PP_ and *ε*_FB_, our model reproduced the intense F–F interactions in short distances (less than 1 Mbps), and strong P–P interactions in long distances, respectively [19]. Due to different interaction profiles between F and P domains, the decay patterns of contact probability for F and P domains differ from each other. Although the α value is not sensitive to *ε*_FF_ and *ε*_PP_, P (F) domains contact more frequently with the increase in these parameters, indicating more condensed P domains and F domains and a higher segregation level between P and F domains. Therefore, the current modeling study provides further support for the role of F-P domain segregation in chromatin structure formation [19]. With development and senescence, the heterochromatin and repressive histone modifications accumulate along the genomic sequence [40,41,42,43], increasing the strength of interactions among P domains (*ε*_PP_ in our model) and with proteins such as HP1α mediating interactions among the heterochromatin [44]. As predicted in our model, the phase-separation extent increases with the increase in *ε*_PP_, which is in line with our previous observations that chromatin compartmentalization increases with the development and senescence of the cell [19].

The spatial organization of the F and P domains is affected by the complex interplay among *ε*_LC_, LAD pattern, *ε*_PP_, and *ε*_FF_. We investigated the effects of the interactions between the polymer beads and the container wall (*ε*_LC_), which are used in the model to mimic the interactions between chromatin and the nuclear envelope. It was reported that the heterochromatin (compartment B) tends to populate near the nuclear periphery and near the nuclear center, while the euchromatin (compartment A) tends to populate in between [1,3,12]. By adjusting the interaction strength between the chromatin and the nuclear envelope, we successfully reproduced this experimental observation. Due to their stronger inter-block interactions and weaker interactions to form TADs, compared to F domains, P domains tend to condense into a globule and reside in the container center when *ε*_LC_ is small. However, a large *ε*_LC_ can disrupt this spatial organization pattern, and serve as a force to pull P domains to the periphery. P domains without this interaction tend to remain near the nuclear center, and such a tendency is enhanced when large *ε*_PP_ and/or small *ε*_FF_ are used. These results imply the important role of LADs in forming the spatial distribution and organization of chromatin. Consistently, it was reported that the heterochromatin of rod cells, which lack lamin, tends to populate in the nuclear center [13,14]. In addition, knockout of lamin could result in the dislocation of heterochromatin [45,46]. Recent experimental studies proposed that the competition between the lamin B1, which tethers heterochromatin to the nuclear periphery, and the nuclear matrix, which pulls the heterochromatin to the nuclear interior, regulates the 3D chromatin structure [47], which is also consistent with our simulation results. Such an effect of lamin was also investigated in some recent simulation studies [3,33,34]. The three-layered spatial organization can be observed in Refs. [3,34]. However, to study the effects of LAD formation, a plane (Refs. [33,34]) or the inner surface of a sphere (Ref. [3] and in the current study) is usually used to mimic the nuclear membrane. While the confinement effects of the nucleus are ignored with a plane, the usage of a sphere tends to overestimate the confinement effects when a smaller spherical container than the real nucleus is used in the model. Besides, the small sphere has a higher ratio between the surface and the volume than the real nuclear. To avoid these effects due to the plane and the small container, a whole-genome model with a container, whose size is comparable to a real nucleus, is desired.

During interphase, chromatins tend to segregate to form chromosome territories, which result in significantly higher intra-chromatin contact frequencies than inter-chromatin ones [12]. The positions of different chromatins are highly correlated with the content of the chromatin: CG-rich chromatins tend to populate inside the nucleus while CG-poor chromatins tend to populate near the nuclear periphery [48]. Our model system including chr18 and chr19 reproduced this relationship between chromatin positioning and genomic contents. In our model, since P domains are mostly C/G poor and colocalize with LADs, they show a higher tendency to populate near the periphery, and as a result, chromatins dominated by P domains such as chr18 show an obvious preference for the nuclear periphery, while chr19, with few P domains, is characterized by an interior positioning. However, different from experimental observations, chromatins in our model are only moderately separated, characterized by the slightly lower inter-chromatin contact frequencies compared to long-range intra-chromatin contact frequencies. These results might be relevant to the attractive nature of the domain interactions used in our model. In addition, there are studies suggesting that the chromosome territories form due to structural memory effects (kinetic effects) of unknotted, topologically-constrained and long polymers [49]. It was reported that the chromatin condensation begins from forming small globules locally [50] and during which, intra-chromosome topological constraints might be established. Our model with relatively short chains and no topological constraints is expected to equilibrate fast without extensively exhibiting the memory effects.

To understand the general and common folding principles of chromatins, our model only takes the sequence properties into consideration, as an indicator of the TAD property, the compartment boundary and the LAD boundary. It is interesting that our simple model has largely reproduced the overall properties of chromatin folding. However, though the sequence plays the most fundamental role, it is only part of the story and chromatin folding exhibits significant cell specificity. For example, compartment boundaries can vary from cell to cell [19]. When parameters based on cell specificity are added, we expect the model can be used to investigate the chromatin structure variation between cells, and obtain a better understanding on the roles of sequence and cell-specific factors in affecting chromatin organization and function.

## 4. Materials and Methods

To establish the coarse-grained model of chromatin, we divided the chromatin into segments of 100 kbps and represented each segment by one bead. If more than 50% base pairs of one segment belong to forests (prairies), the bead corresponding to this segment was assigned to the F domains (P domains), so that the chromatin was coarse-grained into a block copolymer with alternating blocks of F and P domains. If most beads in a domain (more than 50%) were detected with signals of lamina association (samples of single-cell lamina-associating data reported in Ref. [21] were used), the whole domain was recognized as an LAD. Due to the lack of sequence information, we omitted the centromere regions in our model. If there are TAD boundaries in one segment (the data of TAD boundaries were adapted from Ref. [35]), the corresponding bead was recognized as a TAD boundary.

Firstly, we applied a finite extensible nonlinear elastic (FENE) bonding potential between adjacent beads to form a polymer with the form
(1)$$U_{\mathrm{FENE}}\left(r\right)=\{\begin{array}{cc} -0.5KR_{0}^{2}\ln\left[1-{(\frac{r}{R_{0}})}^{2}\right] & r<R_{0}\\ +\infty & r\geq R_{0} \end{array}$$
with $R_{0}$ set to 1.5*σ*_0_ and *K* set to 30.0$\varepsilon_{0}/\sigma_{0}^{2}$, where *σ*_0_ is the length unit corresponding to the diameter of one bead and *ε*_0_ is the energy unit corresponding to *k*_B_*T*_0_*,* which is about 2.6 kJ/mol when *T*_0_ = 308 K (the body temperature). To evaluate the value of *σ*_0_, we simply treated one 100-kbp segment as a 500-nucleosome random-walk chain, and calculated the mean square gyration radius (~75 nm). Therefore, in our model *σ*_0_ is taken as 150 nm. Secondly, we applied a Lennard-Jones (LJ) potential with the form
(2)$$U_{\mathrm{LJ}}\left(r\right)=\{\begin{array}{cc} 4\varepsilon_{\mathrm{FF}/\mathrm{PP}/\mathrm{FP}}\left[\frac{\sigma_{0}^{12}}{r^{12}}-\frac{\sigma_{0}^{6}}{r^{6}}-\frac{\sigma_{0}^{12}}{r_{c}^{12}}+\frac{\sigma_{0}^{6}}{r_{c}^{6}}\right]\; & r<r_{c}\\ 0 & r\geq r_{c} \end{array}$$
to every non-bonding pair of beads (Figure 1a). We choose the cutoff $r_{c}=1.3\sigma_{0}$ because we assume that the interactions between beads are established only when they are contacted. Additionally, the values of $\varepsilon_{\mathrm{FF}/\mathrm{PP}/\mathrm{FP}}$ vary from 3.0*ε*_0_ to 5.0*ε*_0_, to make the energy difference, *U*_LJ_(*r*_c_)–*U*_LJ_(2^1/6^*σ*_0_), in the range of 1.0*ε*_0_~2.0*ε*_0_, a scale corresponding to the Van der Wall’s interactions and have been widely used in previous studies, e.g., Refs. [25,26]. In Equation (1), *ε*_PP_ is set to be larger than *ε*_FF_ considering the higher contact frequency between segments in compartments B. The F–P interactions (*ε*_FP_) are the weakest among the three. Thirdly, we adopted a harmonic bonding potential between adjacent TAD boundaries with the form $\varepsilon =\;\varepsilon_{\mathrm{FB}/\mathrm{PB}}r^{2}$ to mimic the formation of TADs, which are loop-like structures. $\varepsilon_{\mathrm{FB}/\mathrm{PB}}$ is set weak (less than 0.5$\varepsilon_{0}$) as the bonding between TAD boundaries is not permanent. Since interactions inside TADs observed in F domains are generally stronger than those in P domains [8,19], stronger interactions are employed between boundaries of F-TADs than those of P-TADs, therefore *ε*_FB_ > *ε*_PB_. Finally, the model chromatin is placed into a spherical container with a volume 20 times that of the chromatin, calculated from the total volume of nucleosomes and the volume of the cell nucleus. Interactions between LADs and the container (*ε*_LC_) are described with a LJ potential cutoff 1.3*σ*_0_. We assume LADs could bind to the nuclear envelop relatively tightly. Thus, we assigned the value of *ε*_LC_ in the range between 1.5*ε*_PP/FF/FP_ and 2.0*ε*_PP/FF/FP_, which is 6.0*ε*_0_~8.0*ε*_0_. The interactions between other beads and the nuclear envelope are described by an LJ potential cutoff $2^{1/6}\sigma_{0}$.

We used the open-source LAMMPS code (version: 7 Aug 2019) [51] to run our simulations. We initiated the simulation by annealing a random self-avoiding chain by decreasing the temperature from 5.0*T*_0_ to 1.0*T*_0_ gradually, and ran the simulation for $2.0\times \;10^{8}$ steps with a 0.01*τ*_0_ time step corresponding to about 10 ns under 1.0*T*_0_ to avoid exceeding 1/10 of the time scale of the motion with the highest frequency in this system, and therefore one trajectory corresponds to about 40 min in the real world. Each simulation was repeated 4 times from independent initial conformations and velocities. The conformation was saved every 10,000 steps for further analysis.

Two beads are considered to be in contact if their center-to-center distance is less than 2.5*σ*_0_, and the contact probability between two beads was calculated by counting the frequency for them to contact in an ensemble. By averaging the contact probability of the beads with the same sequential distance, we obtained plots of the variation of contact probability (*P*) as a function of the sequential distance (*N*). To compare the simulation results with experiments, we obtained the plots of the contact probability as a function of genomic distance using data adapted from Refs. [35,36], and aligned these plots so that *P* at *N* = 300 kbps in each plot equals to the simulation value.

To calculate the density distribution along the radius of the container, we divided the container into 200 shells with the same thickness, and calculated the density of F and P beads in each shell. The density values obtained from all conformations are averaged to yield the RDF for F and P domains, respectively. We used a relative radial position (normalized radius) as the abscissa of the RDF plots, say *r*/*R* with *R* representing the radius of the container.

## 5. Conclusions

In conclusion, we constructed a block copolymer model which takes into account loop formation, interactions among different polymer domains, and interactions between the polymer and the container, to simulate the chromatin in a cell nucleus. Using this simple model, we were able to reproduce several features of chromatin folding that are observed at various length scales, including the TAD formation, compartmentalization, the power-law decay property of the contact probability as a function of genomic distance, different contact behaviors between compartments A and B, spatial distribution, and the formation of chromosome territories. In this model, loop formation results in the TAD-like structure in the contact map and compartmentalization mainly arises from the mosaic properties of the genome content. These interactions result in the complex decay pattern of contact probability as a function of the linear genomic distance. The loop formation affects both the decay pattern at the scale of TADs and between several Mbps to 20 Mbps. The probability decay pattern is weakly affected by F–F and P–P interactions, though the segregation tendency of F domains and P domains is enhanced with the increase in these interactions. Due to a strong tendency to form TADs in F domains and strong inter-domain P–P interactions, the contact probability inside compartments A is higher than that inside compartments B, while the contact probability between compartments A is lower than that between compartments B. The simulation results suggest that LAD formation is an important driving force in forming different spatial distribution patterns of different compartments. The LADs tend to populate near the nuclear membrane, while P domains not forming LADs tend to condense near the nuclear center. This condensation is also affected by interactions inside P and F domains. Partly due to their difference in genomic contents such as F-P and LAD distributions, different chromatins exhibit different spatial distribution patterns, with chr18 and chr19 being two typical examples. However, segregation between chromatins in our model is limited and the formation of chromosome territories is only partially reproduced in our model. Therefore, more efforts are needed to reveal the mechanism, including both thermodynamic and kinetic factors, behind the chromosome territory formation. We note here, that this study focuses on general properties of chromatin folding given the genomic information which is common to all cells of the same species, cell-specific properties such as epigenetics should be included in the future studies. In addition, the nucleus is a complex and dynamical system, in which many environmental factors, such as temperature [52], mechanical forces/pressure, binding of proteins, and, especially, non-equilibrium effects, are expected to play important roles. The interplay between these complex factors is all worth careful analysis and study. With the improved understanding of the different roles of and interplay between sequence and environments, such a model might be useful to predict environmental effects by adding related parameters, which might be beneficial in understanding the mechanism of many diseases, such as cancer and neurodegenerative disorders.

## Figures and Tables

**Figure 1 ijms-22-01328-f001:**
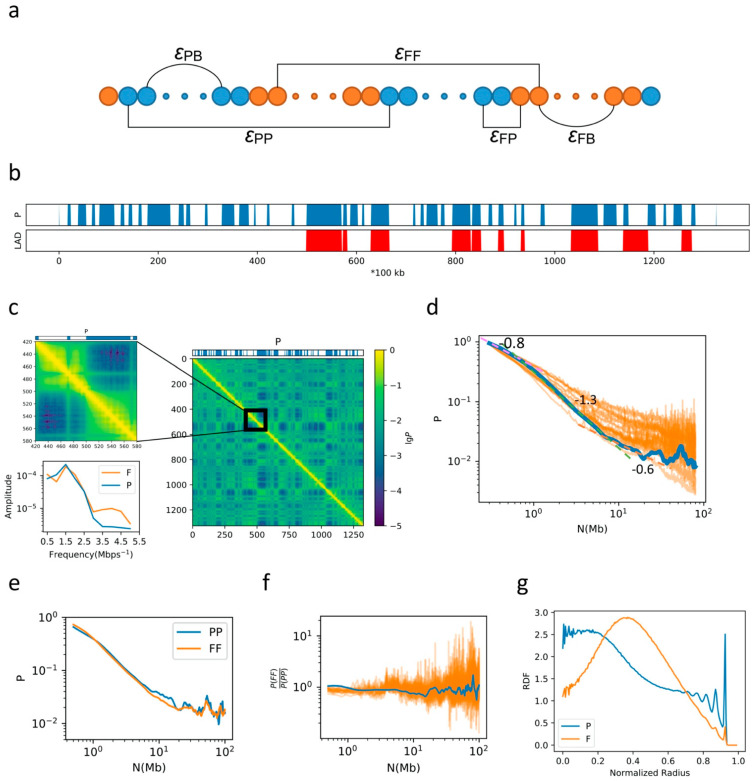
(**a**) A graphical illustration of the model, in which blue beads representing P domains and orange beads representing F domains; (**b**) The P-domain and lamina-associating domains (LAD) annotation of chr10; (**c**) The contact map generated from the simulation and the profile of P domains, and the lower left panal shows the results of windowed Fourier transform; (**d**) The blue plot is the contact probability as a function of sequential distance calculated from the contact map in (**c**), and the orange plots are experimental data from up to 20 cell types; (**e**) The contact probability of F and P domains as a function of sequential distance; (**f**) The ratio between the contact probability of F and P domains, with the blue line calculated from simulation results and orange lines calculated from experimental data; (**g**) The radial distribution for beads in F and P domains.

**Figure 2 ijms-22-01328-f002:**
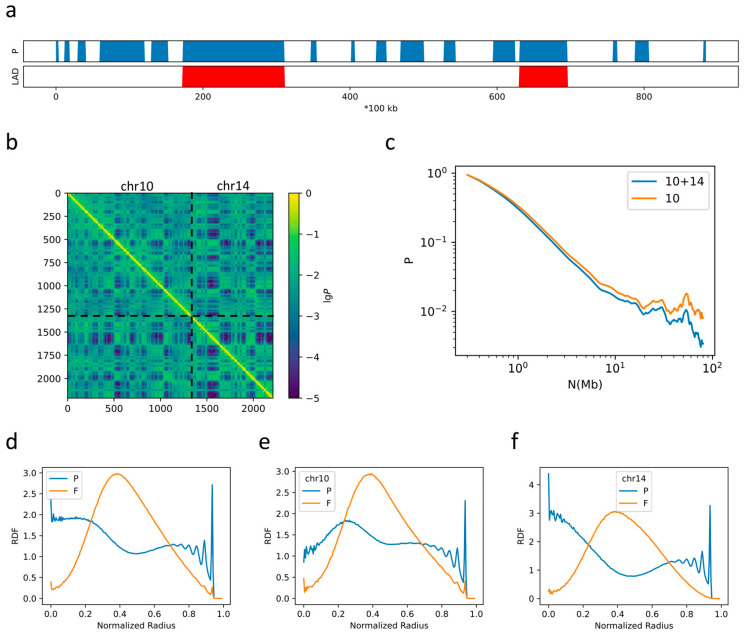
(**a**) The P-domain and LAD annotation of chr14; (**b**) The contact map generated from simulations on chr10 and chr14; (**c**) the average contact probability as a function of the sequential distance; (**d**) the radial distribution functions (RDF) obtained for chr10+chr14 simulation; (**e**) the RDF of Chr10 from the 2-Chr simulation; (**f**) the RDF of Chr 14 in the 2-Chr system.

**Figure 3 ijms-22-01328-f003:**
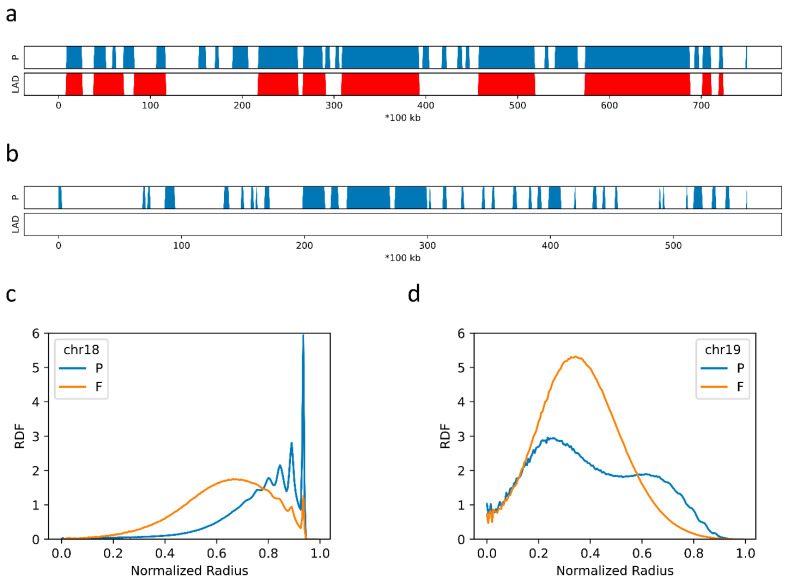
(**a**) The P-domain and LAD annotation of chr18; (**b**) The P-domain and LAD annotation of chr19; (**c**) the RDF obtained from the 3-chr simulation with only chr18 taken into account; (**d**) the RDF obtained from the 3-chr simulation with only chr19 taken into account.

**Figure 4 ijms-22-01328-f004:**
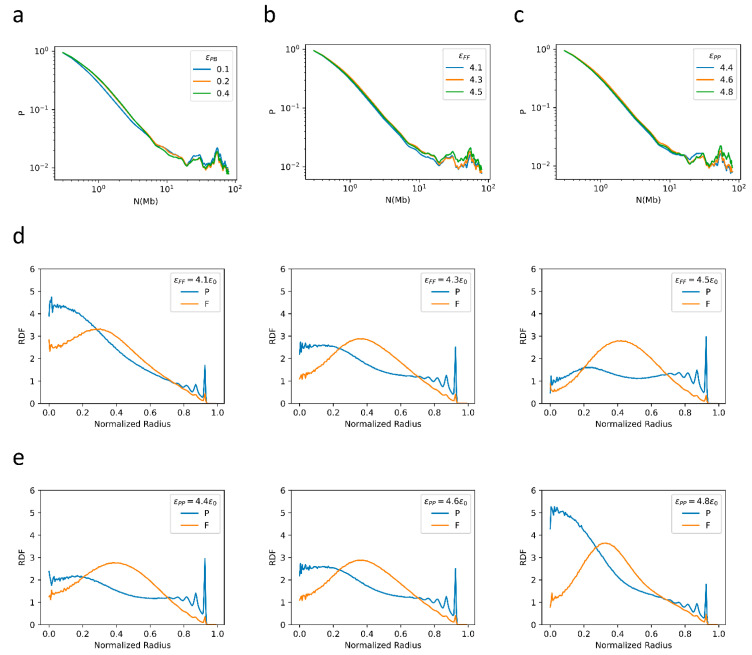
(**a**) the contact probability generated from the 1-Chr system with different *ε*_PB_ values as a function of sequential distance; (**b**) the contact probability generated from the 1-Chr system with different *ε*_FF_ values as a function of sequential distance; (**c**) the contact probability generated from the 1-Chr system with different *ε*_PP_ values as a function of sequential distance; (**d**) the RDFs obtained from the 1-Chr system with different *ε*_FF_ values; (**e**) the RDFs obtained from the 1-Chr system with different *ε*_PP_ values.

**Figure 5 ijms-22-01328-f005:**
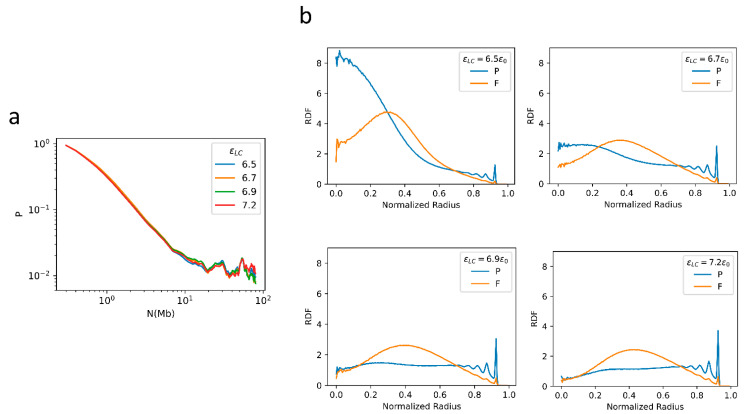
(**a**) The contact probability as a function of sequential distance generated from the 1-Chr system with different *ε*_LC_ values; (**b**) the RDFs obtained from the chr10 simulation with different *ε*_LC_ values.

**Figure 6 ijms-22-01328-f006:**
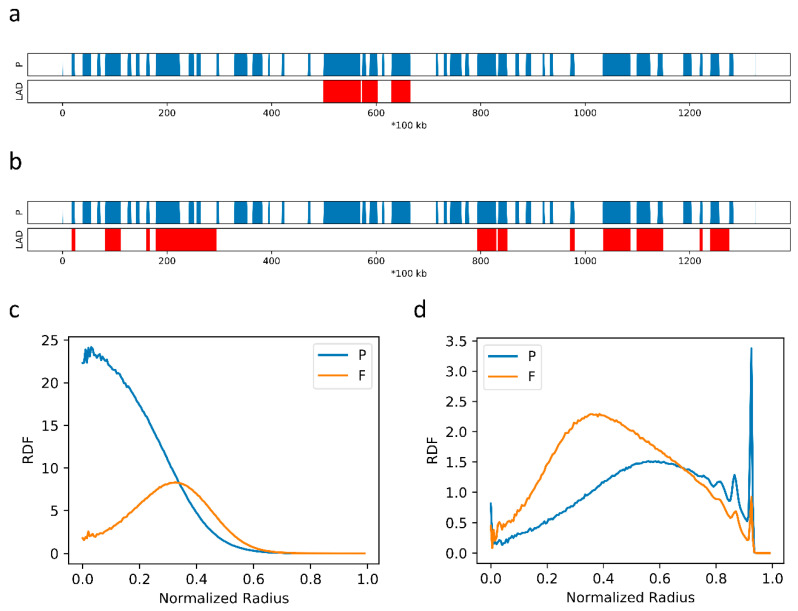
(**a**,**b**) The P-domain and LAD annotation of chr10 from two single-cell samples; (**c**) the RDFs obtained from the chr10 simulation of samples in Figure 6a; (**d**) the RDFs obtained from the chr10 simulation of samples in Figure 6b.

## Data Availability

The datasets generated and/or analyzed during this study are available from the corresponding authors on reasonable request.

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
