# Peer review of "A DNA Sequence Based Polymer Model for Chromatin Folding"

_ijms, 2021, doi:10.3390/ijms22031328_

Round 1
Reviewer 1 Report
This manuscript investigates a block copolymer model of genome folding based on genomic annotation.
After a generic introduction on chromosome nuclear organization and polymer modeling, authors described their polymer model. Then, they showed a set of simulation on chromosome 10 alone, followed by a set on chromosome 10+14 and finally on chromosome 14+18+19. Finally, they concluded and discussed their results.
In the recent years, chromosome organization became a major field of investigation in modern biology and polymer modeling is now routinely used to investigate mechanistic hypotheses on the driving forces of genome folding. Therefore, this work is potentially interesting. However, I have major overall concerns:
- The approach used by the authors is not at all original and has been developed and used with a much higher degree of sophistication previously by many other groups.
- The model, choice of parameters and simulations are not clearly explained or motivated.
- Results are not properly quantified or compared to existing experimental, and often lead to overstatements by the authors. Moreover, results are not original and mainly confirm predictions (and corresponding biological interpretations) already made by other groups using similar models.
And major specific issues:
- In addition to the many works existing on block copolymer (Jost, Di Pierro, Brackley, etc.), 3 papers combining block copolymer + LAD-like interaction should be more precisely discussed in the introduction and the discussion sections (Falk et al, Nature 2019; Wang et al, NAR 2019; Sati et al, Mol Cell 2020).
- Introduction L75..78: “However, the models so far tend to concentrate … at different scales.” This is not true. Many polymer models describe genome folding at multi-scales, describing the hierarchical chromosome organization from TADs to territories (Nuebler et al & many other works by Mirny’s group, Di Pierro et al and many other works by Onuchic/Wolynes groups, Brackley et al or Buckle et al and many works by Marenduzzo groups, Shi et al and many works by Thirumalai/Hyeon groups, Sati et al, Di Stefano et al and many other works by Jost’s group, Chiariello et al and many other works from Nicodemi’s groups, etc.).
- The model is really badly described in the Results but also in the Method parts:
- A brief description (coarse-graining, type of interactions, confinement, etc.) of the polymer model at the beginning of the Results section would be welcomed. This may help understanding rapidly what epsilon_PP, epsilon_FF, .. represent without reading the Method part
- The choice of parameter values is not motivated. In general, the rationale behind choice of interaction potentials, the identity of the chromosomes or the parameters is not very clear.
- For the Lennard-Jones, authors used a cut-off distance of 1.3 sigma_0 which is very weak, why? More standard values are between 1.6 and 2.5.
- Why using short-range potentials (Lennard-Jones) for PP/FF/PF/.. interactions and a long-range spring-like potential for BE?
- Details on the number of independent simulated trajectories should be given. How long in terms of real units (in sec or min) one simulated trajectory correspond to?
- It seems that initial configurations are random, equilibrated conformations of a random-walk. Many authors (Rosa & Everaers, PLos CB 2008 for example) underline the importance of starting from mitotic-like chromosome conformation since long polymers will keep a partial memory of the initial state, in particular for long-range (beyond 20 Mbp) interactions. Can authors comment?
- From what I understood, F & P are derived from analysis of the genomic sequence. Does it mean that the F & P 1D partitionings are independent of the cell type? However, TADs (even if many of them are conserved) and above all compartments are cell-type specific and mainly depend on epigenomic features (CTCF, histone marks, etc.). LAD also can be cell-type specific (at least the so-called variable LADs). Can the authors comment on that?
- Many times in the text, the authors claim that their predictions are in quantitative agreement with experimental observations. However, the paper does not contain any quantitative comparisons with Hi-C data (which are freely available). Predicted Hi-C maps and contact probabilities P can (and should) be compared to experimental data in order to see if the proposed model is relevant. For example, it’s clear than large-scale organization (intra-chromosome interactions > 20 Mbp or inter-chromosome interactions or chromosome territories) are not captured by the simulations.
- Authors simulate one chromosome (or two or three) inside a sphere. In this case, the ratio between the inner surface of the sphere and its volume (~1/Radius) is much higher than in the actual nucleus (with 46 chromosomes) since the Radius of the confining sphere will be lower. This clearly will favor surface (LAD) and confinement effect. Can the authors comment on the relevance of this type of simulations for investigating the radial distribution function (RDF) of a chromosome inside the nucleus?
- Analysis of the spatial occupation using chi^2 is unclear. Comparisons with values of chi^2 computed for a null model (no specific interactions) may help.
More minor issues:
- In the introduction, L32-.., ‘alpha’ is missing at many positions along the text.
- 1C: information on the P, F and LAD status of the genomic regions shown in the HiC would help making the parallel between the 1D annotation and the 3D folding.
- Introduction L 82: by ‘chromatin’, I guess the authors refer to ‘chromosome’?
- L93: ‘BE’ is not defined before.
- The ‘Materials and Methods’ section is located between the Discussion and Conclusion. It is quite an original position for a methodological section.
- What are the parameters of the FENE used by the authors (a mathematical expression of a FENE potential may help also)?
- L278 on the choice of LAD: why only one-sample was used?
- Eps_0, sigma_0 are not defined nor estimated.
- The authors are not modeling the nucleolus or any specific interactions with it, so why evoking the nucleolus at many positions in the text (L120 for example).
- P, F & LAD profiles should be ginve for Chr 14.
- Fig 3A: what is MCS?
- Discussion L211: it is strange to place on the same level a mechanistic model (loop-extrusion) and a phenomenological model (self-returning).
- Discussion L222: “which was less noticed…”. Not true, many other studies already discuss that (ok maybe more in a A/B or epigenomic context rather than P/F).
Author Response
Response to Reviewer 1 Comments
This manuscript investigates a block copolymer model of genome folding based on genomic annotation.
After a generic introduction on chromosome nuclear organization and polymer modeling, authors described their polymer model. Then, they showed a set of simulation on chromosome 10 alone, followed by a set on chromosome 10+14 and finally on chromosome 14+18+19. Finally, they concluded and discussed their results.
In the recent years, chromosome organization became a major field of investigation in modern biology and polymer modeling is now routinely used to investigate mechanistic hypotheses on the driving forces of genome folding. Therefore, this work is potentially interesting. However, I have major overall concerns:
The approach used by the authors is not at all original and has been developed and used with a much higher degree of sophistication previously by many other groups.
The model, choice of parameters and simulations are not clearly explained or motivated.
Results are not properly quantified or compared to existing experimental, and often lead to overstatements by the authors.
Moreover, results are not original and mainly confirm predictions (and corresponding biological interpretations) already made by other groups using similar models.
Response: We thank the reviewer for these useful comments. We totally agree with the reviewer that there have been a large number of sophisticated models on chromatin structure, in particular on reproducing the Hi-C contact matrix. Some of these studies are referred to in the manuscript and a more complete list was given in a recent review article (Zhou et al., 2020, PCCP). Though many models have been developed, some important questions remain in understanding the overall principle of chromatin folding, such as the sequence dependence. To expand the feasibility of the model, it is still necessary to integrate these models, and discuss the interplay between parameters. Recently, many studies have emerged following this manner, one representative study of which is made by Nuebler et al mentioned by the reviewer, which integrates the loop extrusion model and compartmentalization. For example, the model by Nuebler used a periodic boundary and thus did not explain the different spatial organization of compartments A and B. The model reported in “Falk et al, Nature, 2019” integrated the block copolymer model and LAD formation, but properties such as the decay of contact probability as a function of genomic distance was not discussed.
The main aim of our study is in fact to examine how a simple model that based mainly on the DNA sequence, but not other complicate information such as epigenetics, can account for the overall chromatin organization in differentiated cells. Therefore, our model is built on DNA sequence. Such a model is motivated by the experimental observation that although chromatin organization varies from cell to cell, but different (especially differentiated cells) share many common (and to a large extent conserved) features such as TADs and compartments. From our earlier analysis it was found that the compartment (TAD) boundaries (determined by 3D structure) overlap nicely with the boundaries between two different DNA domains (determined by the 1D sequence). These two linear domains are distinguished by their different densities of CpG islands (CGI). They also differ significantly in gene distribution and epigenetic marks, with the domains of higher CGI density are rich in genes, marked by active epigenetic modifications and tend to form compartment A.
We found that this simple model did reproduce important features of chromatin organization, in particular, the decay pattern of the contact probability over a large range of linear genomic distances. We did realize and make it clearer in the revised manuscript that although the genomic sequence plays a fundamental role, it cannot account for all details of chromatin folding. Significant variations do exist between different cell types, since the fate of a cell is also significantly affected by epigenetic modifications. Our aim is thus to use the sequential data to reproduce general patterns of chromatin folding, including formation of TADs and compartments, decay of the contact probability with the genomic distance, different spatial organization patterns of compartments A and B, and different intra- and inter-domain interaction profiles of compartments A and B, but not to compare the simulation result with the Hi-C data pixel by pixel (although we did test how the contact map can be affected by various parameters, including the distribution and size of LADs).
We have modified our statements in the revised version to make clear what our model is aimed at. We have also added a discussion on these limitations of our model in the “discussion” section.
And major specific issues:
In addition to the many works existing on block copolymer (Jost, Di Pierro, Brackley, etc.), 3 papers combining block copolymer + LAD-like interaction should be more precisely discussed in the introduction and the discussion sections (Falk et al, Nature 2019; Wang et al, NAR 2019; Sati et al, Mol Cell 2020).
Response: We have added discussions in both the introduction section (L93-L109: “In recent studies, the block copolymer model and LAD formation have been combined to illustrate the spatial organization of chromatin. These models generally divide the chromatin into different kinds of blocks. Blocks of the same kind attract each other. For some blocks, attractions between blocks and nuclear periphery are applied to mimic the LAD formation. For example, M. Falk et al. considered three kinds of blocks, one kind of which could interact with the nuclear lamina [3]. Their results indicate that the LAD formation plays a crucial role in forming the spatial organization pattern for both normal cells and rod cells. In another study, Sati et al. used this kind of model to study the formation of senescence-associated heterochromatin foci. They indicated that these foci form due to loss of interactions between senescence-associated heterochromatins and the nuclear lamina, and weakening in interactions between senescence-associated heterochromatin domains. [35] Although significant variations do exist between different cell types, different (especially differentiated) cells share many common (and to a large extent conserved) features such as those in TAD [4, 5] and compartment formation [20]. A model that provides a simple understanding of the general folding properties of chromatin such as formation of TADs and compartments, decay of the contact probability with the genomic distance, different spatial organization patterns of compartments A and B, and different intra- and inter-domain interaction profiles of compartments A and B, and in particular, the role of DNA sequence, is still in need..”) and the discussion section (L329-L333: “Such an effect of lamin was also treated in some recent simulation studies (refs. [3] and [35]). The model in ref. [3] treated LAD and other heterochromatin as two kinds of blocks to get the three-layered spatial organization of chromatin and model in ref. [35] resulted in the formation of outer heterochromatin and inner euchromatin, but not the three-layered spatial organization of the chromatin.”). Unfortunately, we were not able to find the paper in NAR published in 2019 (neither in recent years) by Wang et al on this topic. We might have missed something here.
Introduction L75..78: “However, the models so far tend to concentrate … at different scales.” This is not true. Many polymer models describe genome folding at multi-scales, describing the hierarchical chromosome organization from TADs to territories (Nuebler et al & many other works by Mirny’s group, Di Pierro et al and many other works by Onuchic/Wolynes groups, Brackley et al or Buckle et al and many works by Marenduzzo groups, Shi et al and many works by Thirumalai/Hyeon groups, Sati et al, Di Stefano et al and many other works by Jost’s group, Chiariello et al and many other works from Nicodemi’s groups, etc.).
Response: We are thankful for the reviewer to the comment. We have modified this part of discussion to describe better these models (L102-109: “Although significant variations do exist between different cell types, different (especially differentiated) cells share many common (and to a large extent conserved) features such as those in TAD [4, 5] and compartment formation [20]. A model that provides a simple understanding of the general folding properties of chromatin such as formation of TADs and compartments, decay of the contact probability with the genomic distance, different spatial organization patterns of compartments A and B, and different intra- and inter-domain interaction profiles of compartments A and B, and in particular, the role of DNA sequence, is still in need.”).
The model is really badly described in the Results but also in the Method parts:
A brief description (coarse-graining, type of interactions, confinement, etc.) of the polymer model at the beginning of the Results section would be welcomed. This may help understanding rapidly what epsilon_PP, epsilon_FF, .. represent without reading the Method part
Response: we have added a brief description at the beginning of the “result” section [L125-L132: “In our model, one chromosome is coarse-grained into a polymer with each bead representing 100 kbps. The information about F/P domains, TAD boundaries and LADs is then mapped to each bead. We applied attractive Lennard-Jones (LJ) potentials between each P-P, F-F and P-F bead pair of different intensity, marked as εPP, εFF, and εPF respectively. To form loops between adjacent TAD boundaries and for simplicity, we applied different harmonic potentials (εPB and εFB) according to the genomic content of the TAD. The model chain is then placed into a spherical container representing the nucleus. Between LADs and the nuclear envelope, an attractive LJ potential of intensity εLC is applied”].
The choice of parameter values is not motivated. In general, the rationale behind choice of interaction potentials, the identity of the chromosomes or the parameters is not very clear.
Response: we have added explanations in the “materials and methods” section of the revised version, including the explanation on values of εFF/PP/FP (L383-385: “Besides, we consider the LJ potential generates an energy difference at the same scale of the Van der Waal’s force, and therefore we choose the value of to 3.0ε0~5.0ε0.”) and εLC (L395-396: “We assume LADs could bind to the nuclear envelop relatively tightly. Thus, we assigned the value of εLC in the range between 1.5εPP/FF/FP and 2.0εPP/FF/FP, which is 6.0ε0~8.0ε0.”).
For the Lennard-Jones, authors used a cut-off distance of 1.3 sigma_0 which is very weak, why? More standard values are between 1.6 and 2.5.
Response: Firstly, we used εFF/PP/FP around 4.0ε0. Though cutoff 1.3σ0 is short, these values generate an energy difference about 1.5ε0, corresponding to a LJ potential cutoff 2.5σ0 with εFF/PP/FP~1.5ε0. It is not a weak interaction. Secondly, though cutoff values 1.6σ0~2.5σ0 used mostly, there are studies using cutoff cutoff 1.3σ0 (McGovern et al, Soft Material, 2016) or 1.5σ0 (Panico et al, 2010, Modelling Simul. Mater. Sci. Eng). Here, we used a cutoff 1.3σ0 because we assumed that only when two beads are close enough (~ 10 nm), the interactions between them are established. We have added the explanations on such a choice in the “materials and methods” section (L382: “We choose the cutoff 1.3σ0 because we assume that the interactions between beads are established only when they get contacted.”).
Why using short-range potentials (Lennard-Jones) for PP/FF/PF/.. interactions and a long-range spring-like potential for BE?
Response: PP/FF/PF interactions are not specific interactions, which arise from different properties of P and F, and one common choice is the LJ potential. However, interactions between BEs are intended to form loops to simulate TAD formation, a spring-like potential is more proper, which has been used in many other studies (such as Bohn et al., PLOS ONE 2010; Fudenberg et al., Cell Reports, 2016).
Details on the number of independent simulated trajectories should be given. How long in terms of real units (in sec or min) one simulated trajectory correspond to?
Response: we have added these details in the “materials and methods” section of the revised version (L400-402: “…corresponding to about 5 ns under 1.0T0, and therefore one trajectory corresponds to about 20 min in the real world. Each simulation was repeated 4 times from independent initial conformations and velocities.”).
It seems that initial configurations are random, equilibrated conformations of a random-walk. Many authors (Rosa & Everaers, PLos CB 2008 for example) underline the importance of starting from mitotic-like chromosome conformation since long polymers will keep a partial memory of the initial state, in particular for long-range (beyond 20 Mbp) interactions. Can authors comment?
Response: As stated in our manuscripts, our model could only partially reproduce the formation of chromosome territories. Actually, these studies on partial memory of the initial state have motivated us to attempt to reproduce the formation of chromosome territories by using two chains with quite a long distance as initial inputs. However, we found that the chains approach each other and mix quickly, and the results show no significant difference with a system started from a random, equilibrated initial conformation. It appears that a partial memory of the initial state might be affected by the detailed setup of the model.
From what I understood, F & P are derived from analysis of the genomic sequence. Does it mean that the F & P 1D partitionings are independent of the cell type? However, TADs (even if many of them are conserved) and above all compartments are cell-type specific and mainly depend on epigenomic features (CTCF, histone marks, etc.). LAD also can be cell-type specific (at least the so-called variable LADs). Can the authors comment on that?
Response: We totally agree with the reviewer’s comments. As stated earlier in the response, our main purpose in this paper to examine how DNA sequence itself can be used to understand the general features of chromatin organization. As pointed out by the reviewer, there is significant variation in chromatin structure formation between different cell types. But various studies also showed that great similarity is shared between them, which deserves an understanding on how the common property they all share, the sequence, affects the folding of chromatin. It is true that F & P are derived from genomic sequence, and they are independent of the cell type. The observation that a simple model based on DNA sequence can reproduce general features of the HiC data supports that the DNA sequence does play an important role in chromatin structure formation. However, the chromatin folding is not only determined by the genomic sequence. Other factors such as environments result in different cell fate, including epigenetic marks, and chromatin 3D structure. Although it is not our focus here, the lack of consideration on cell specificity is a limitation of our study, and we have clarified this point in the “discussion” section (L355-L363: “To understand the general and common folding principles of chromatins, our model takes only the sequence properties into consideration, as an indicator of the TAD property, the compartment boundary and the LAD boundary. It is interesting that our simple model has largely reproduced the overall properties of chromatin folding. However, though the sequence plays the most fundamental role, it is only part of the story and chromatin folding exhibits significant cell specificity. For example, compartment boundaries can vary from cell to cell [20]. When parameters based on cell specificity are added, we expect the model can be used to investigate the chromatin structure variation between cells, and obtain a better understanding on the roles of sequence and cell-specific factors in affecting chromatin organization and function.”).
Many times in the text, the authors claim that their predictions are in quantitative agreement with experimental observations. However, the paper does not contain any quantitative comparisons with Hi-C data (which are freely available). Predicted Hi-C maps and contact probabilities P can (and should) be compared to experimental data in order to see if the proposed model is relevant. For example, it’s clear than large-scale organization (intra-chromosome interactions > 20 Mbp or inter-chromosome interactions or chromosome territories) are not captured by the simulations.
Response: We have made clear that quantitative comparison is for general features such as the decay slope at different length scales, which as stated in the previous now made clear that our model mainly fits for the scale up to 20 Mbps in both the “result” (L149) and the “discussion” section (L282). As we have mentioned above, the genomic sequence is a fundamental player but not the only one in chromatin folding. Therefore, using simulations on a simple sequence-based model, we try to understand why many different cells tend to show similar general properties of chromatin folding, although they are of different fates and are characterized by different epigenetic marks and different populations of structural proteins such as CTCF and cohesins. Therefore, our model based on genomic is used to reproduce general patterns of chromatin folding, but not the Hi-C data in detail. For the scale larger than 20 Mbps, there is a sharper decay in contact probability (Lieberman-Aiden et al., Science, 2009). It seems that this pattern is relevant to inter-chromatin interactions, as shown in Figure 2c. We agree with the reviewer that since our model does not reproduce fully the formation of chromosome territories, and therefore we did not focus on distances above 20M.
Authors simulate one chromosome (or two or three) inside a sphere. In this case, the ratio between the inner surface of the sphere and its volume (~1/Radius) is much higher than in the actual nucleus (with 46 chromosomes) since the Radius of the confining sphere will be lower. This clearly will favor surface (LAD) and confinement effect. Can the authors comment on the relevance of this type of simulations for investigating the radial distribution function (RDF) of a chromosome inside the nucleus?
Response: We agree with the reviewer that the small nuclear could result in a high ratio between surface and volume. If this is the dominant factor, one would expect that the P domains in the 2-chr system should show higher tendency to populate near the nuclear center compared to the 1-chr system, as the container radius of the 2-chr system is 15% larger than 1-chr system. However, the P domains in the 2-chr system show a higher tendency to populate near the nuclear periphery (Figure 2d and figure 1f). This result indicates that the inter- and intra-chromosome interactions play more important roles than the effects of the size of container. We did include a short discussion on this possible effect (L333-338: “When one more chromatin is added to the system, the density of P domains in the nuclear center decreases, though the P domains in the 2-Chr system should show higher tendency to populate near the center as a large nucleus could result in a high ratio between the surface and the volume of the container. Therefore, the inter- and intra-chromosome interactions play more important roles than the effects of the size of container in our model.”).
Analysis of the spatial occupation using chi^2 is unclear. Comparisons with values of chi^2 computed for a null model (no specific interactions) may help.
Response: We have deleted this analysis and now abandoned the clustering method to describe the spatial occupation.
More minor issues:
In the introduction, L32-.., ‘alpha’ is missing at many positions along the text.
Response: We have fixed these problems.
1C: information on the P, F and LAD status of the genomic regions shown in the Hi-C would help making the parallel between the 1D annotation and the 3D folding.
Response: This is a very good suggestion. We have added this profile to Figure 1c.
Introduction L 82: by ‘chromatin’, I guess the authors refer to ‘chromosome’?
Response: we have fixed these statements.
L93: ‘BE’ is not defined before.
Response: we have replaced such a statement by “TAD boundary”.
The ‘Materials and Methods’ section is located between the Discussion and Conclusion. It is quite an original position for a methodological section.
Response: We followed the order provided in the original template.
What are the parameters of the FENE used by the authors (a mathematical expression of a FENE potential may help also)?
Response: we have added the mathematical expression of the FENE potential (L374).
L278 on the choice of LAD: why only one-sample was used?
Response: We have added related data and discussions in the “result” section (L265-L272: “To evaluate the effects of LAD pattern, we finally considered two more LAD samples of chr10, the profiles of which are shown in Figures 6a and 6b. The LADs in Figure 6a are of relative low population and concentrated, while the LADs in Figure 6b are more abundant and dispersed. We concentrated mainly on the RDF. As shown in Figure 6c, the P domains of the sample in Figure 6a condense into the nuclear center. In contrast, the P domains of the sample in Figure 6b are drawn to the periphery (Figure 6d), and neither P nor F domains exhibit a preference for the central location. These results indicate the abundance and distribution of LADs affect strongly the spatial organization of the interphase chromatin.” and Figure 6) and the “discussion” section (L312).
Eps_0, sigma_0 are not defined nor estimated.
Response: we have estimated these values in the revised version. Epsilon_0 corresponds to about 2.6 kJ/mol (L377) and sigma_0 corresponds to about 70 nm (L377).
The authors are not modeling the nucleolus or any specific interactions with it, so why evoking the nucleolus at many positions in the text (L120 for example).
Response: we have changed this statement with “the nuclear center”.
P, F & LAD profiles should be given for Chr 14.
Response: we have added these profiles in Figure 2a.
Fig 3A: what is MCS?
Response: It is the Monte Carlo step. However, this figure has been deleted in the revised version.
Discussion L211: it is strange to place on the same level a mechanistic model (loop-extrusion) and a phenomenological model (self-returning).
Response: We tried to emphasize that the slow decay at the TAD scale has drawn attention widely from very different aspects. We deleted the word “theoretical” in front of this sentence.
Discussion L222: “which was less noticed…”. Not true, many other studies already discuss that (ok maybe more in a A/B or epigenomic context rather than P/F).
Response: As far as we know, different interactions inside compartments are relatively less noticed. However, we have deleted these potentially misleading statements.
Reviewer 2 Report
In this study focused that DNA-sequence based coarse-grained polymer model considering different inter- and intra-domain interactions, interactions of loop forming, as well as interactions between chromatin and container. This model captures the various single-chromatin properties and partially reproduces the formation of chromosome territories. The presentation of this good work however needs more significant improvement. Since, the MS needed some more innovations of the study.
- Abstract: Abstract is not clear. Please improve the abstract section.
- Introduction: Background of the study should be made to very clear. Provide more details of introduction and review of the work.
- Please speculate about the reasons for the obtained results. Discussion need to improve.
- Conclusion: the authors should add the significance of this research to potential practical application.
- The whole of MS needs to be improved. English writing and paragraph needs to be improved.
Author Response
Response to Reviewer 2 Comments
In this study focused that DNA-sequence based coarse-grained polymer model considering different inter- and intra-domain interactions, interactions of loop forming, as well as interactions between chromatin and container. This model captures the various single-chromatin properties and partially reproduces the formation of chromosome territories. The presentation of this good work however needs more significant improvement. Since, the MS needed some more innovations of the study.
Abstract: Abstract is not clear. Please improve the abstract section.
Response: We thank the reviewer for suggestions on the abstract. We clarify the relationship between the CGI density and spatial domains in the revised version (L19-L21: “These spatial domains are found to highly correlate with the mosaic CpG island (CGI) density. High CGI density corresponds to compartments A and small TADs, and vice versa.”), and make the description of the model clearer (L24-26).
Introduction: Background of the study should be made to very clear. Provide more details of introduction and review of the work.
Response: We thank the reviewer for suggestions on the introduction. We added an introduction on the TAD in the revised version (L35-37: “At the scale of hundreds of kilo-base pairs (kbps), the topological associating domains (TADs) have been revealed, which are contiguous loop-like structures along the sequence. The interactions inside TADs are stronger than between TADs.”), and different TAD types in different compartments (L43-44: “the TADs in compartments B are large and lack strong interactions inside, while the TADs in compartments A are small and condensed”). More illustrations on LADs have been added in the revised version (L72-74: “Lamin proteins populate mainly near the inner-surface of the nucleus (the nuclear lamina), and the LADs are sequences interacting with these proteins. Single cell experiments showed that not all LADs bind to the nuclear lamina.”). Besides, we added more descriptions on the models we reviewed in the “introduction” (L85-102: “Many characteristics of chromatin folding have been reproduced or revealed from these rather sophisticated model studies. For example, the string and binder switch (SBS) model of Barbieri et al. used free beads to mediate interactions between chromatin segments, and reproduced the scaling properties of the entire chromatin, topological domain formation, as well as looping out [28]. The loop extrusion model described a process that protein complexes slide along the chromatin, which can be halted by TAD boundaries [12,33]. This latter model explains well the formation of TADs. Furthermore, by combining the loop extrusion model with a block copolymer model, J. Nuebler et al. studied the interplay between the formation of TADs and compartments. [34] In recent studies, the block copolymer model and LAD formation have been combined to illustrate the spatial organization of chromatin. These models generally divide the chromatin into different kinds of blocks. Blocks of the same kind attract each other. For some blocks, attractions between blocks and nuclear periphery are applied to mimic the LAD formation. For example, M. Falk et al. considered three kinds of blocks, one kind of which could interact with the nuclear lamina [3]. Their results indicate that the LAD formation plays a crucial role in forming the spatial organization pattern for both normal cells and rod cells. In another study, Sati et al. used this kind of model to study the formation of senescence-associated heterochromatin foci. They indicated that these foci form due to loss of interactions between senescence-associated heterochromatins and the nuclear lamina, and weakening in interactions between senescence-associated heterochromatin domains. [35]”).
Please speculate about the reasons for the obtained results. Discussion need to improve.
Response: We added a discussion on the reason for the slow-fast-slow decay pattern (L285-286: “At the short range, our simulation showed that the slow decay is due to the formation of TADs.” and L292-295: “In our model, the fast and then slow decay at distances longer than the TAD size appear to be resulted from the heterogenous nature of the block copolymer. For a block copolymer with blocks of equal length, a periodically oscillating contact probability is observed as a function of sequential distance [38].”), and in the remaining problems of fully reproducing the chromosome territory (L350: “Such results might originate from the attractive nature of the domains in our model.”) in the discussion section.
Conclusion: the authors should add the significance of this research to potential practical application.
Response: This is really a valuable suggestion. We believe such a model based on sequence could help us to clarify the effects of sequence and environments respectively. Potentially, these efforts might help us to better understand mechanisms of many diseases such as cancer and neurodegenerative disorder. We have added these potential applications into the “Conclusion” section (L445-448: “With the improved understanding of the different roles of and interplay between sequence and environments, such a model might be useful to predict environmental effects by adding related parameters, which might be beneficial in understanding the mechanism of many diseases like cancer and neurodegenerative disorder.”).
The whole of MS needs to be improved. English writing and paragraph needs to be improved.
Response: We have fixed grammar mistakes and tried to improve the writing. Besides, we added subtitles in the “result” section to make the paragraphs better organized.
Round 2
Reviewer 1 Report
The authors have made extensive editing of the manuscript that clearly improved the quality of the paper. I think there is still two major issues that were not clearly addressed by the authors and that need to be fixed before acceptance.
1) Comparison to experimental data: it should be relatively easy to plot the average contact frequency P as a function of the genomic distance (as well as the contact probabilities of F and P domains) from Hi-C data of several human cell types, in order to (at least visually) compare the slope and the overall shape of these curves with predictions.
2) Motivations of the choice of the parameters is still unclear: why 70nm for sigma_0, why 5ns for the time-scale, why setting energies at the scale of VdW forces? (we are talking about a coarse-grained model with interactions between 100 kbp beads! not real molecular interactions), etc.
Please find below more minor points related to the changes or responses provided by the authors in their revision:
" the discussion section (L329-L333: “Such an effect of lamin was also treated in some recent simulation studies (refs. [3] and [35]). The model in ref. [3] treated LAD and other heterochromatin as two kinds of blocks to get the three-layered spatial organization of chromatin and model in ref. [35] resulted in the formation of outer heterochromatin and inner euchromatin, but not the three-layered spatial organization of the chromatin.”).
Response: in Ref. [35], they do observe a mild three layered spatial organization (it's just that they do not mention it explicitly), see Sup Fig 3E.
"Unfortunately, we were not able to find the paper in NAR published in 2019 (neither in recent years) by Wang et al on this topic. We might have missed something here."
Response: my mistake, the reference is Chiang et al, Cell Report 2019, and it should be discussed as well.
"PP/FF/PF interactions are not specific interactions, which arise from different properties of P and F, and one common choice is the LJ potential. However, interactions between BEs are intended to form loops to simulate TAD formation, a spring-like potential is more proper, which has been used in many other studies (such as Bohn et al., PLOS ONE 2010; Fudenberg et al., Cell Reports, 2016)."
Response: well transient loops may certainly arise from LJ potential...Note that in Fudenberg et al, they use short-range soft-core potentials to 'force' strong loops...
"As stated in our manuscripts, our model could only partially reproduce the formation of chromosome territories. Actually, these studies on partial memory of the initial state have motivated us to attempt to reproduce the formation of chromosome territories by using two chains with quite a long distance as initial inputs. However, we found that the chains approach each other and mix quickly, and the results show no significant difference with a system started from a random, equilibrated initial conformation. It appears that a partial memory of the initial state might be affected by the detailed setup of the model"
Response: to have the formation of chromosome territories 'for free' as in Rosa & Everaers, the underlying polymer model should be an unknotted, topologically-constraint, long polymer. Any coarse-grained model that want to maintain this physical property need to be properly designed (see also Ghosh & Jost, PLoS CB 2018). Clearly, here the used coarse-graining model will lead to fast equilibrium and hence to loose the structural memory (ie, chromosome territory formation) emerging from topological constraints.
"We agree with the reviewer that the small nuclear could result in a high ratio between surface and volume. If this is the dominant factor, one would expect that the P domains in the 2-chr system should show higher tendency to populate near the nuclear center compared to the 1-chr system, as the container radius of the 2-chr system is 15% larger than 1-chr system. However, the P domains in the 2-chr system show a higher tendency to populate near the nuclear periphery (Figure 2d and figure 1f). This result indicates that the inter- and intra-chromosome interactions play more important roles than the effects of the size of container. "
Response: Well it's not so straightforward. Increasing confinement effect will clearly enhance the bulk phase (P phase more in the center) while increasing surface effect will promote the localization at the periphery. On top of that, chr10 and chr14 have not the same length, nor the same lad composition...
Other minor issues:
L64: "chromatin"-> "chromosome"
Fig1a: the figure is absent
Fig1c, left panel: please add the P domains
L154: precise the size of the spherical shell used to confined chr10 (same for 2Ch or 3Chr cases)
L161-162: "different contact patterns are observed..." : please quantify
L198: "profile"->"profiles", "is"->"are"
Fig4a: "eps_PC"->"eps_PB"
L268 "the contact" -> "the average contact"
L277: you may cite Falk, Sati & Chiang papers
L381: "chromatin"->"chromosome"
L400 "Such results might originate...": not clear that this is a valid origin
L402-404: the relation between the first and second part of the sentence if unclear.
L.430: this is not a FENE potential (missing the log)
L.431 looking at the formula in L430, the unit of K is not an energy...
L.435: it should be (sigma_0/r)^12 in the formula, the last part of the formula has also mistakes, please correct.
Author Response
The authors have made extensive editing of the manuscript that clearly improved the quality of the paper. I think there is still two major issues that were not clearly addressed by the authors and that need to be fixed before acceptance.
1)Comparison to experimental data: it should be relatively easy to plot the average contact frequency P as a function of the genomic distance (as well as the contact probabilities of F and P domains) from Hi-C data of several human cell types, in order to (at least visually) compare the slope and the overall shape of these curves with predictions.
Response: We thank the reviewer for these good suggestions. We have added related comparisons in Figure 1d and Figure 1f, and we have also included related discussions (L162: The ratio between the contact frequency for F and P domains as a function of genomic distance is now compared with experimental data (Figure 1f), showing a reasonable agreement except for a slight overestimation at the TAD scale.)
2) Motivations of the choice of the parameters is still unclear: why 70nm for sigma_0, why 5ns for the time-scale, why setting energies at the scale of VdW forces? (we are talking about a coarse-grained model with interactions between 100 kbp beads! not real molecular interactions), etc.
Response: We have added related discussions (L391: “To evaluate the value of σ0, we simply treated one 100-kbp segment as a 500-nucleosome random-walk chain, and calculated the mean square gyration radius (~ 35 nm). Therefore, in our model σ0 is taken as 70 nm.”, L417: “to avoid exceeding 1/10 of the time scale of the motion with the highest frequency in this system” and L398: “Besides, values of εPP/FF/FP vary from 3.0ε0 to 5.0ε0, to make the energy difference, ULJ(rc)-ULJ(21/6σ0), in the range of 1.0ε0~2.0ε0, a scale corresponding to the Van der Wall’s interaction and widely used in previous studies, e.g., refs. [27] and [28]” ).
Please find below more minor points related to the changes or responses provided by the authors in their revision:
" the discussion section (L329-L333: “Such an effect of lamin was also treated in some recent simulation studies (refs. [3] and [35]). The model in ref. [3] treated LAD and other heterochromatin as two kinds of blocks to get the three-layered spatial organization of chromatin and model in ref. [35] resulted in the formation of outer heterochromatin and inner euchromatin, but not the three-layered spatial organization of the chromatin.”).
Response: in Ref. [35], they do observe a mild three layered spatial organization (it's just that they do not mention it explicitly), see Sup Fig 3E.
Response: We have modified the discussion in L340-346: “Such an effect of lamin was also investigated in some recent simulation studies [3,51,35]. The three-layered spatial organization can be observed in Refs. [3] and [35].”
"Unfortunately, we were not able to find the paper in NAR published in 2019 (neither in recent years) by Wang et al on this topic. We might have missed something here."
Response: my mistake, the reference is Chiang et al, Cell Report 2019, and it should be discussed as well.
Response: we have added related discussions in L99-103: “M. Chiang et al. [51] and S. Sati et al. [35] used this type of model to study the formation of senescence-associated heterochromatin foci (SAHF). Though the criteria they used to divide blocks are different, their studies both indicate the important roles of the interactions between related heterochromatin domains and the nuclear lamina, as well as interactions between heterochromatin domains, in the formation of SAHF.”
"PP/FF/PF interactions are not specific interactions, which arise from different properties of P and F, and one common choice is the LJ potential. However, interactions between BEs are intended to form loops to simulate TAD formation, a spring-like potential is more proper, which has been used in many other studies (such as Bohn et al., PLOS ONE 2010; Fudenberg et al., Cell Reports, 2016)."
Response: well transient loops may certainly arise from LJ potential...Note that in Fudenberg et al, they use short-range soft-core potentials to 'force' strong loops...
Response: We agree with the reviewer that transient loops may arise from LJ potential, as seen in the literature (for example, Brackley et al., 2016, NAR). However, using non-specific LJ potential may establish interactions between TAD boundaries of large genomic distances (thus across many F/P domains or between different chromosomes). Here, we try to simulate the formation of relatively stable loops only between adjacent TAD boundaries, which can be conveniently achieved by applying a harmonic potential to assigned bead pairs.
We thank the reviewer for pointing out our mis-citation of Fudenberg’s study. Sanborn et al. 2015 PNAS did use the harmonic potential to form loops.
"As stated in our manuscripts, our model could only partially reproduce the formation of chromosome territories. Actually, these studies on partial memory of the initial state have motivated us to attempt to reproduce the formation of chromosome territories by using two chains with quite a long distance as initial inputs. However, we found that the chains approach each other and mix quickly, and the results show no significant difference with a system started from a random, equilibrated initial conformation. It appears that a partial memory of the initial state might be affected by the detailed setup of the model"
Response: to have the formation of chromosome territories 'for free' as in Rosa & Everaers, the underlying polymer model should be an unknotted, topologically-constraint, long polymer. Any coarse-grained model that want to maintain this physical property need to be properly designed (see also Ghosh & Jost, PLoS CB 2018). Clearly, here the used coarse-graining model will lead to fast equilibrium and hence to loose the structural memory (ie, chromosome territory formation) emerging from topological constraints.
Response: We thank the reviewer for these good explanations. We have added related discussions in L360: “In addition, there are studies suggesting that the chromosome territories form due to structural memory effects (kinetic effects) of unknotted, topologically-constrained and long polymers [52]. It was reported that the chromatin condensation begins from forming small globules locally [48] and during which intra-chromosome topological constraints might be established. Our model with relatively short chains and no topological constraints is expected to equilibrate fast without exhibiting extensively the memory effects.”
"We agree with the reviewer that the small nuclear could result in a high ratio between surface and volume. If this is the dominant factor, one would expect that the P domains in the 2-chr system should show higher tendency to populate near the nuclear center compared to the 1-chr system, as the container radius of the 2-chr system is 15% larger than 1-chr system. However, the P domains in the 2-chr system show a higher tendency to populate near the nuclear periphery (Figure 2d and figure 1f). This result indicates that the inter- and intra-chromosome interactions play more important roles than the effects of the size of container. "
Response: Well it's not so straightforward. Increasing confinement effect will clearly enhance the bulk phase (P phase more in the center) while increasing surface effect will promote the localization at the periphery. On top of that, chr10 and chr14 have not the same length, nor the same lad composition...
Response: We thank the reviewer for these analyses in depth. We have modified the corresponding text in L340-347: “However, to study the effects of LAD formation, a plane (Refs. [35] and [51]) or the inner surface of a sphere (Ref. [3] and in the current study) is usually used to mimic the nuclear membrane. While the confinement effects of the nuclear are ignored with a plane, the usage of a sphere tends to overestimate the confinement effects when a smaller spherical container than the real nuclear is used in the model. Besides, the small sphere has a higher ratio between the surface and the volume than the real nuclear. To avoid these effects due to the plane and the small container, a whole-genome model with a container, whose size is comparable to a real nuclear, is desired.”
Other minor issues:
L64: "chromatin"-> "chromosome"
Response: We have modified the corresponding terms.
Fig1a: the figure is absent
Response: The absence could be due to a display issue. We did include the figure.
Fig1c, left panel: please add the P domains
Response: We have added the P domains in this panel.
L154: precise the size of the spherical shell used to confined chr10 (same for 2Ch or 3Chr cases)
Response: We have added information on the size of these spherical shells (L133: “The model chain is then placed into a spherical container representing the nucleus, and the chromosome occupies a volume fraction φ=5% of the container volume.”, and L139: “In this model, the radius of the container is 16.7σ0”, and L184: “the radius of the container is changed to 19.8σ0 to maintain φ=5%” and L213: “in which all parameters are the same as those used in the 2-Chr model as the amounts of beads are almost identical in these two models.”)
L161-162: "different contact patterns are observed..." : please quantify
Response: We have quantified these patterns using a windowed Fourier transform as shown in Figure 1c and L145-149: “To quantify these different patterns, we applied a windowed Fourier transform to the 5th diagonal of the contact matrix, with a window size of 2 Mbps. The results of Fourier transform for windows containing mainly F and P domains are averaged respectively, and shown in Figure 1c.”.
L198: "profile"->"profiles", "is"->"are"
Response: We have fixed these statements.
Fig4a: "eps_PC"->"eps_PB"
Response: We have made the change.
L268 "the contact" -> "the average contact"
Response: We have changed this term in subsection 2.2.
L277: you may cite Falk, Sati & Chiang papers
Response: We have cited these papers, in L269 “which is also noted in ref. [3], [51] and [35]”.
L381: "chromatin"->"chromosome"
Response: We have fixed this statement.
L400 "Such results might originate...": not clear that this is a valid origin
Response: We have modified these statements and added a related discussion on the memory effects as mentioned previously.
L402-404: the relation between the first and second part of the sentence if unclear.
Response: we have modified this sentence (L363-365: “It was reported that the chromatin condensation begins from forming small globules locally [48] and during which intra-chromosome topological constraints might be established. Our model with relatively short chains and no topological constraints is expected to equilibrate fast without exhibiting extensively the memory effects.”).
L.430: this is not a FENE potential (missing the log)
Response: We have made correction on this issue.
L.431 looking at the formula in L430, the unit of K is not an energy...
Response: We have corrected this error.
L.435: it should be (sigma_0/r)^12 in the formula, the last part of the formula has also mistakes, please correct.
Response: We have corrected these errors.
Reviewer 2 Report
Requested corrections were completed.
Author Response
We thank the reviewer for these helpful comments.
Round 3
Reviewer 1 Report
Authors have addressed most of my concerns. However, Fig 1 (where many changes have been done) is still badly displayed (see attached file) : panel A with the model is not visible, the orange curve (experimental curves) on panel D and F are not present. I would like to see before making my final decision.
minor issue: references 49 & 52 are the same

Author Response
Authors have addressed most of my concerns. However, Fig 1 (where many changes have been done) is still badly displayed (see attached file) : panel A with the model is not visible, the orange curve (experimental curves) on panel D and F are not present. I would like to see before making my final decision.
Response: The absence is due to a display issue, which might result from different versions of Microsoft Office. We do include these figures. We have added a snapshot of Fig. 1 in the attached file.
minor issue: references 49 & 52 are the same
Response: We have corrected this error.
